# Factors Influencing Work Ability among the Working-Age Population in Singburi Province, Thailand

**DOI:** 10.3390/ijerph19105935

**Published:** 2022-05-13

**Authors:** Teeraphun Kaewdok, Saowanee Norkaew, Sanpatchaya Sirisawasd, Nattagorn Choochouy

**Affiliations:** 1Faculty of Public Health, Rangsit Campus, Thammasat University, Pathum Thani 12121, Thailand; saowanee.n@fph.tu.ac.th (S.N.); sanpatchaya@fph.tu.ac.th (S.S.); 2Research Unit in Occupational Ergonomics, Thammasat University, Pathum Thani 12121, Thailand; nattagorn.c@fph.tu.ac.th; 3Faculty of Public Health, Lampang Campus, Thammasat University, Lampang 52190, Thailand

**Keywords:** work ability, occupational health, safety, working population, health status

## Abstract

The ability to work is an essential factor in the quality of work life. This study aimed to determine factors related to work ability among the working population in Sing Buri Province. A total of 360 workers completed a cross-sectional survey using a questionnaire. The questionnaire included demographic data and work conditions, psychosocial factors, and measures of the seven components of the work-ability index (WAI). Mean, percentage, standard deviation, and stepwise multiple regression analyses were conducted to determine the rates and predictors of the WAI of the participants. Of the 360 participants, 61.40% were women with an average age of 43.00 ± 11.30 years; 36.70% had completed high school education. Their average work experience was 11.30 ± 8.50 years. The results revealed that the average WAI was 31.40 ± 4.15. 76.10% of the participants qualified for the moderate work ability index. The number of current diseases, age, and monthly income were found to significantly contribute to the prediction of the WAI (*p* < 0.05). This finding suggests that the relevant agencies should support a policy, project or program related to health promotion among the working population with physical health conditions. Promoting occupations should be considered to raise income policy. In doing so, work ability among the working population may be improved.

## 1. Introduction

In Thailand, the working-age group is the main population in the country. In 2019, the National Statistical Office of Thailand reported that more than 60% of the employment rate or 57 million came from the labor market. In terms of the labor force among the Association of Southeast Asian Nations (ASEAN), East Asia and the Pacific region countries, it was found that around 38 million reported in the fourth- and sixth-largest countries, respectively [1], related to the economic expansion increase in industry and agriculture and services sectors. Nowadays, elder workers conflict with the labor market due to health status, care responsibilities, and education levels [2]. Since an aging population developing quickly in Thailand, these challenges are concerned. In 2060, the size of the elderly population, i.e., those aged more than 65, is predicted to rise from 13% today to 31% [2]. This trend is consistent with a previous study by Fischer et al., (2021) [3], Nygaard et al., (2021) [4], and Rodríguez and colleagues (2021) [5], who concluded that the workforce is aging in many countries around the world. There is strong evidence demonstrating that fast changes in the nature of work and technology will require workers with new skills to fill labor market needs created by population aging [2]. Both hard and soft skills are needed; an excellent technical understanding of emerging digital technologies and communication, critical thinking, and persuasion are required [2,6]. Healthy workers and decent work refer to the need for aspirations for people in their working lives, relating to protecting labor rights, promoting safe and secure working environments [3].

The health needs of workers comprise the physical fitness and working ability needed to perform any task. Work ability is defined as a person’s addressing capability with a given job at a given time and is considered one of the essential factors of work [5,7]. It is believed that working with high work ability can achieve their work with high-quality performance. Meanwhile, the incongruence between an individual’s ability and job requirements could deteriorate their working performance and health status [5,8,9,10]. Health effects and intense workloads may result in an unbalance between the individuals’ physical, intellectual resources and the emotional, cognitive, and physical demands and characteristics of their work, thus leading to a decline in their work ability [11]. Previous studies demonstrated possible factors affecting WAI. Several studies mentioned that occupational conditions are related to workers’ health status [10,12].

Many factors influence WA, including the individual’s physical attributes (e.g., age, weight and musculoskeletal capacity) and job-related tasks (e.g., physical workload, mental work demands, decision authority, skill discretion, supervisor support, social support and meaning of work). Other factors such as lifestyle (e.g., leisure-time physical activity, diet, smoking and sleep) [13,14,15] and work environment (i.e., chemical hazards, psychological and ergonomics) [16] may also affect WA. Previous studies found that those hazards lead to health problems, including cardiovascular disorder, stress, mental health, respiratory symptoms, musculoskeletal disorder, and accidents [6,8,17,18,19,20]. Thus, balancing work ability and work demand is a concern due to health effects, work capability and quality of life [16]. The Work-Ability Index (WAI) is a tool used to measure the ability of the worker to fit into their job. It focuses on an individual’s resources concerning the work demands [21]. This tool allows for a self-evaluation of individual factors and factors associated with the performance of a specific type of work, affecting work ability to varying degrees [6].

Developing countries are experiencing significant demographic changes as people live longer and have healthier lives, particularly in Thailand. The working age is essential in driving and developing the country in prosperity, society, economy, and well-being. The labor force comprises people ages 15 and older who supply labor to produce goods and services during a recent period. Work ability among workers or employees is vital for working practice quality and constantly changing working life. As a result, several sectors in the national authorities are trying to promote work ability throughout the working life, which expect everybody will stay longer in the labor market. These campaigns include considering changes in the legal age for retirement and preparing their citizens [18]. Thailand has a career-promotion system for those of working age and the elderly, and efforts are being made to extend the voluntary working life of the elderly. Various projects are in place to encourage employers to retain and hire the elderly [1]. Understanding factors affecting work ability for the working population is necessary to promote and maintain good work ability in all phases of working life [5]. Several previous studies have shown that the differences between the local context in terms of socioeconomic and population structure may be affected by WAI, such as demographic data, the occupational sector, psychosocial factors, and other factors. Thus, the study results may be helpful for relevant agencies in occupational health management to support workers’ efficiency among the target group. This study’s results may support Thailand’s sustained economy through the sustainable development goals (SDGs) and occupational health perspective [7,8]. 

Based on background and rationale, the hypothesis of this study was concerned with the demographic characteristics and psychosocial factors influencing work ability among the working population. Therefore, this study aimed to explore factors influencing work ability among workers in Singburi Province, Thailand. The study’s findings can be used to support policymakers in promoting the work ability of workers in Thailand through occupational health protection, income policy, and health promotion for all.

## 2. Materials and Methods

### 2.1. Study Design

A cross-sectional study was conducted to investigate the work ability level and factors influencing work ability among the working-age population. Job characteristics included current occupational, work experience, and occupational health and safety training. The factors that influence WA include personal factors such as sex, age, body mass index, education background, monthly income, and health conditions. The psychosocial factors were also included in this study. These factors were selected because they are relevant to WAI from a previous study [6,8,16,17,18,19,20].

### 2.2. Study Participants 

The study participants were workers aged between 15 and 60 years living in Promburi districts, Singburi Province, Thailand. Singburi has a total population of 3289 people. The 360 workers were recruited and selected by convenience sampling techniques if the workers resided in the study area and were willing to participate in the study. Participants were calculated using statistical power analysis (GPower Version 3.1.9.2, Mannheim, Germany). Data were collected in May and July 2020. 

### 2.3. Study Tools

This study used a four-part questionnaire that was developed from the literature review. The first part contained general information, including age, gender, education, income, body mass index, work experience, and current occupation. The second part asked about working conditions. The third section included the job content questionnaire (JCQ). The thoroughly recommended JCQ scales of decision latitude, psychological demands, physical job demands, job security, social support, and hazards at work were included in the questionnaire. Each part comprised three items. Four responses are possible for each item: “strongly disagree” (code 1) to “strongly agree” (code 4). The cut-off values for the scale were constructed according to Karasek’s recommendations. Furthermore, they were dichotomized at the median of the total sample of participants for use in the analyses [22]. This study combined psychosocial factors into two categories: low and high. It is consistent with a previous study [23,24]. The version of the JCQ obtained following transcultural adaptation to the Thai context was used [25]. The final part was adapted from the work ability index (WAI), developed by the Finnish Institute of Occupational Health [16]. A questionnaire is a standard tool used worldwide to determine WAI among workers in different occupational sectors. Previously, the questionnaire was used to investigate public health personnel in Spain [5], the older working population in Denmark [4], and the working population in Poland [6]. Meanwhile, the questionnaire was utilized to investigate workers from large and small–medium enterprises [21] and their workers [17], caregivers [11], and older workers [26], respectively, in Cyprus, Brazil, and Thailand. WAI is composed of ten items divided into seven aspects, as follows [8,11,27]:(1)Current work ability compared with best work ability throughout life, where a score of 0 represents complete work disability and a score of 10 represents work ability at its best;(2)Work ability concerning the demands of the job;(3)Number of current diseases diagnosed by a physician;(4)Estimated work impairment due to those illnesses or injuries;(5)Sick leave taken in the past 12 months;(6)The worker’s prognosis of work ability two years from now;(7)The worker’s mental resources.

The range of the summative index was 7 to 49, which is classified into poor (7–27), moderate (28–36), good (37–43), and excellent (44–49) work ability. In the current study, the questionnaire was piloted with a similar group of participants to test reliability by Cronbach’s alpha coefficient, more than 0.6. The well-trained personnel also interviewed the participants during the house visit.

### 2.4. Data Analysis

The general information, job characteristics, working environments, psychosocial factors, and work-ability index were determined by descriptive statistics: frequency, percentage, mean, and standard deviation (SD). Factors influencing work-ability data were determined by stepwise multiple regression analysis at the significant level of 0.05 and 95 percent confidence interval.

## 3. Results

Information from the examination of the participants’ demographics is presented in Table 1. Most of the participants were female (61.4%) and average aged 43 years old (SD = 11.3). The participants have a body mass index (BMI) at a healthy level (1.7%) with an average BMI of 23.8 kg/m^2^. Most of participants working 7.9 h per day (SD = 1.8, min–max = 3–14) and 5.9 days per week (SD = 1.1, min–max = 1–7).

The results of psychosocial factors showed that most of the participants had high job strain in decision latitude (80.3%), job security (74.7%), and social support (88.3%). In comparison, more than half of them had low job strain in psychological demands (65.6%), physical job demands (59.2%), and hazards at work (73.6%). The summary of the results is presented in Table 2.

The average WAI among the participants was 31.4 ± 4.15, with minimal and maximal WAI values of 21.0 and 46.0, respectively. Given the WAI value, 76.1% of participants were classified as the moderate work-ability group, 14.25% had a low ability to work, 6.4% of participants evaluated their ability to work as good, and 3.3% as having an excellent ability to work (Table 3).

Correlation analyses between dependent variables (e.g., age, body mass index, monthly income, working hours per day, working hours per week, and congenital diseases) and independent variable: WAI with multicollinearity test showed no relative correlations higher than the average 0.65 (Table 4). Thus, there was no collinearity among all those analysis factors.

The factors influencing work ability data showed that the number of current diseases, age and monthly income were significantly associated with WAI (*p*-value < 0.001). The standard error of the estimate equals ±3.328, as shown in Table 5.

## 4. Discussion

The ability to work results from the interaction between job requirements in terms of physical and mental strain, capacities and skills of the workers [6]. The study indicated that the average value of WAI was 31.4 and 76.1% of the participants represented to moderate the ability to work. These results are similar to workers in Asian countries in the Philippines (35.6) [28], Malaysia (35.3) [29], and Indonesia (39.6) [30]. Conversely, other studies have reported a slightly higher worker population of 41.1 in Vietnam [30] and 40.1 in China [31]. These results demonstrated that the working population is physically and mentally ready and enthusiastic. Moreover, they enjoy working and living in moderation. It is possible that the working population in recent studies with an average age of 43 years. In line with their educational background, 32.7% had finished high school. Their average work experience was 11.3 years. In addition, most occupations were in the service and industrial sectors. Several occupations are still involved with work that requires more physical and mental attention. These occupations might be at a higher risk due to accidents or occupational illnesses. Juszczyk et al., (2019) [6] concluded that the ability to work is often defined as the relationship between a person’s resources and requirements specific to a particular type of work. Excellent work ability improves employees’ quality of life and well-being. 

Various risk factors may cause the ability to work among workers. According to this study, the number of current diseases, age, and monthly income were found to significantly contribute to the prediction of the WAI among the working population in Rong Chang sub-districts and Promburi districts, Singburi Province. The results of this study concur with the conclusions of previous studies that claimed the ability to work depends on many and varied factors [6,21]. Similarly, studies in China reported that WAI is influenced by individual and work-related factors [31].

This study found that many current diseases were a significant correlation with WAI. It might be explained that many physical conditions are diseases with many complex symptoms. Many factors are involved with treatment, such as food, medicine, rest, and exercise, including daily life and work. Work ability is unsuitable for workers with many underlying diseases. It is possible that for the specific symptoms of these diseases, people commonly complain of intangible symptoms such as pain, fatigue, and mood disorders. In recent studies, the working population had an average age of 43 years old, while the physical work capacity between the ages of 40 and 60 years was reported to decline by 20% due to the decrease in aerobic and musculoskeletal capacity. These declines can decrease the work ability and substantially increase work-related injuries and illness [32]. The present results are consistent with previous studies regarding the effect of the health status on work ability [20]. Following previous studies, the health status is regarded as the most significant impact on work ability among different age groups in Cyprus [21]. A longitudinal study found that chronic health problems were related to lower and decreased WAI in middle-aged Dutch employees [33]. Work ability is multifactorial and reflects the balance between work demands and the worker’s capacity to cope with those demands [4]. Therefore, there is a need to promote and maintain health status for good work ability in all phases of working life.

Recent studies indicated that age was negatively associated with work ability [34,35]. This study also showed that age was found to significantly predict the WAI among the working population with a negative correlation. With the increasing trend of older workers in the workforce, the health needs of older workers comprise physical and mental function, fitness, and the working ability needed to perform the job. This finding, supported by the study by Hirapara et al., (2021) [36], indicated that work ability could be deteriorated with workers’ age, especially in sedentary employment. In southern Thailand, the study of Thanapop et al., (2021) [26] also reported that a low to moderate WAI was more likely among older workers. The improvement of occupational safety, health and environment or health promotion programs could be beneficial in supporting work ability among the working population despite differences in the work setting and educational level [26,37].

Furthermore, if the relationship between work performance and various physical and psychosocial risk factors can be improved, it could help workers to live longer [34]. Therefore, the early detection of risk among workers at an increasing age can enhance their functional capability and productivity [36]. A study of west Ethiopian health workers found that a high monthly income is one factor that encourages workers to be more satisfied with their jobs [38]. Monthly income was found to significantly contribute to WAI prediction among the working population in this study, with a positive correlation. This finding is consistent with Yingratanasuk et al., (2015) [39], who discovered that if workers work for a more extended period, their monthly income increases, demonstrating their high level of job satisfaction. However, to ensure the health and safety of their employees, organizations must strike an appropriate balance between working hours and monthly income.

This study had no significant effect on psychosocial factors regarding job demand control. The literature review in many countries found that psychosocial factors can contribute to work ability, including individual psychological and psychosocial factors in terms of high job demand, low control, and low social support [40,41,42]. The results may be explained by differences in occupational sectors, time pressure, workload, and organizational policy in different countries. Psychosocial job characteristics may contribute to work-related behaviors and burnout among employees [42,43]. These can lead to a decrease in the ability to work. The previous studies revealed significant relationships between psychosocial factors and WAI [17,44]. Ma et al., (2014) [41] mentioned the need for psychosocial support for expatriate workers to improve their work ability and undertake further research related to occupational health. The claim is consistent with a previous study by Phakthongsuk and colleagues (2008) [25], who concluded that several researchers in occupational medicine had raised concerns regarding the adverse effect of the psychosocial work environment on health in Thailand. This study’s findings suggest that WAI promotion should be considered in psychosocial aspects.

There are some limitations in this cross-sectional study that do not permit cause–effect. Limitations of self-reported data can be occurred by recall bias. The findings from this study cannot be generalized to the working population who work in other areas in Thailand since the study considered participants from only one province in central Thailand. It is suggested for future research to propose a longitudinal study together with the recruitment of a large sample size. Further study would also be needed to compare other areas and occupational sectors. Future research directions can be examined at the country level with various occupational and environmental influences. Nevertheless, an essential strength of the present study is that the findings may raise awareness and help promote and maintain good work ability. 

## 5. Conclusions

This study found that the average WAI was moderate. The number of current diseases, age, and monthly income significantly contribute to the WAI prediction. These factors are essential for considering occupational health and safety policy to promote work ability in the working population. This finding may pave the way to improve a policy, project or program related to health promotion among the working population with physical health conditions. Practical policies should promote employees’ participation in order to increase their WAI.

## Figures and Tables

**Table 1 ijerph-19-05935-t001:** Demographic information and socio-demographic of the respondents (*n* = 360).

Demographic Characteristics	Number	Percentage
Gender		
Male	139	38.6
Female	221	61.4
Age group (years old)		
<20	6	1.7
21–30	52	14.4
31–40	96	26.7
41–50	97	26.9
51–60	106	30.3
Mean ± SD: 43.0 ± 11.3		
Weight (Kilograms: kg)		
Mean ± SD: 62.3 ± 12.4		
Height (Centimeters: cm)		
Mean ± SD: 161.7 ± 12.4		
Body Mass Index: BMI (kg/m^2^)		
Mean ± SD: 23.8 ± 4.6		
Less than 18.5 (Underweight)	24	6.7
18.5–22.9 (Healthy)	150	41.7
23.0–24.9 (Overweight)	78	21.7
25.0–29.9 (Obese)	73	20.3
More than 30.0 (Extremely obese)	35	9.7
Education		
Primary school	69	19.2
Secondary School	90	25
High School	132	36.7
Bachelor’s Degree	58	16.1
Postgraduate	11	3.1
Monthly income (Thai Baht)		
Less than 5000	83	23.1
5000–10,000	178	49.4
10,000–15,000	69	19.2
More than 15,000	30	8.3
Work experience (Year)		
1–5	122	33.9
6–10	96	26.7
>10	142	39.4
Mean ± SD: 11.3 ± 8.5		
Currently job		
Agriculture	8	2.2
Employee (private sector)	202	56.1
Government official (public sector)	49	13.6
Merchant (Self-employed)	46	12.8
Others	55	15.3
Congenital disease		
No	260	72.2
Yes	100	27.8
Number of current diseases (*n* = 100)		
1	43	43
2	20	20
3	15	15
4	7	7
≥5	15	15
Occupational Health and Safety training		
Never	91	25.3
Ever	269	74.7

**Table 2 ijerph-19-05935-t002:** Psychosocial factors at work (*n* = 360).

Psychosocial Factors	Number	Percentage
Decision latitude		
Low	71	19.7
High	289	80.3
Psychological demands		
Low	236	65.6
High	124	34.4
Physical job demands		
Low	213	59.2
High	147	40.8
Job security		
Low	91	25.3
High	263	74.7
Social support		
Low	42	11.7
High	318	88.3
Hazard at work		
Low	265	73.6
High	95	26.4

**Table 3 ijerph-19-05935-t003:** Work ability index (WAI) of seven-dimension summative index (*n* = 360).

Work Ability Index	Male *n* (%)	Female *n* (%)	All *n* (%)
Poor	14 (3.90)	37 (10.30)	51 (14.20)
Moderate	111 (30.83)	163 (45.27)	274 (76.10)
Good	11 (3.05)	12 (3.35)	23 (6.40)
Excellent	3 (0.80)	9 (2.50)	12 (3.30)

Note: There is no significant difference in WAI between males and females by the Chi-squared test.

**Table 4 ijerph-19-05935-t004:** Pearson’s correlation analysis for different individual and work-related factors.

Factors	X_1_	X_2_	X_3_	X_4_	X_5_	X_6_	X_7_	X_8_	Y
Age (X_1_)	1.00								
Body mass index (X_2_)	0.12 *	1.00							
Monthly income (X_3_)	−0.13	−0.17 **	1.00						
Number of current diseases (X_4_)	−0.24 **	−0.20 **	0.12 **	1.00					
Work experience (X_5_)	0.35 **	−0.18 *	0.16 **	0.00	1.00				
Working hour per day (X_6_)	−0.13 *	0.02	0.22 **	0.09	0.09	1.00			
Working day per week (X_7_)	0.11	0.02	−0.10 *	0.02	0.26	−0.37 **	1.00		
Congenital diseases (X_8_)	0.30 **	0.27 **	−0.19 **	−0.62 **	0.04	−0.13 *	0.05	1.00	
WAI (Y)	−0.45 **	−0.13 **	0.21 **	0.48 **	−0.11 *	0.11 *	−0.04	−0.37 **	1.00

Note: * *p*-value < 0.05, ** *p*-value < 0.01.

**Table 5 ijerph-19-05935-t005:** Factors associated with WAI among the working population in stepwise multiple regression analysis.

Variables	b	SE_b_	beta	t	*p*-Value
Constant	30.502	1.121		27.207	<0.001
Number of current diseases	0.956	0.109	0.384	8.771	<0.001
Age	−0.129	0.017	−0.340	−7.761	<0.001
Monthly income	0.572	0.207	0.118	2.766	0.006

SE_est_ = ±3.328; R = 0.603; R^2^ = 0.364; F = 67.886; *p*-value < 0.001.

## Data Availability

The data presented in this study are available on request from the corresponding author.

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
