# Peer review of "Factors Influencing Work Ability among the Working-Age Population in Singburi Province, Thailand"

_ijerph, 2022, doi:10.3390/ijerph19105935_

Round 1

Reviewer 1 Report

I appreciated the opportunity to read your research entitled “ Factors Influencing Work Ability among the Population in Singburi Province, Thailand”. I applaud the attempt to demonstrate work ability among individuals in Thailand's context. However, I have several minor concerns and recommendations about the paper; 

  1. Authors should modify the manuscript title and make it more catchy. For example, ‘Factors influencing work ability of (generation x y z): a case of Southeast Asian country’. With such modification, one’s publication can reach scholars working in different areas (i.e., marketing, HR/OB, supply chain, health & occupational psychology, etc.). 
  2. It’s not always incongruence, but performance expectations also deteriorate performance and lead to counterproductivity. Other than this, the psychological climate has also an impact on one’s work ability and work-related behaviours. Follow the below-given articles.
  3. Authors must emphasize more on contribution part – the last paragraph of the introduction. A contextual gap is not enough for SCIE/Scopus indexed journals. 
  4. How did authors deal with the common method bias of self-reported data collection? 
  5. How did authors approach participants using convenient sampling and how much time did they take to collect data from all 360? Was there any missing data? 
  6. The majority of respondents are female. Briefing on the comparison between both genders and their work ability can also strengthen the contribution current manuscript. 
  7. Line 226 – This study had no significant effect on psychosocial factors regarding job demand control. The authors should provide further justification for this result. Recommended reads and citations; https://doi.org/10.23749/mdl.v110i4.8025 ï¼›https://doi.org/10.2478/manment-2019-0057ï¼›https://doi.org/10.3389/fpsyg.2016.01082 ï¼›https://doi.org/10.1037/10173-008 ï¼›https://doi.org/10.2147/PRBM.S34256ï¼›
  8. Limitations and future directions should also be extended. 

I hope my comments will help you to improve the content. Good luck! 

Author Response

Dear editor,

Please find enclosed our revised manuscript. We thank the two reviewers for their comments. We have revised the manuscript accordingly and provide specific answers in red highlight for reviewer 1 and green highlight for reviewer 2 

Reviewer 2 Report

Dear authors, thank you for the opportunity to get acquainted with the results of your research, which is very interesting and valuable.
To better present the research results, you need to make changes to the following sections of the article:
1. Introduction: despite the fact that the introduction provides data on the relevance of this study, there is not enough information about what factors affect work ability  (i.e., the rationale for why the authors chose these factors in their study is not sufficiently provided).
2. Study design. It is necessary to present in more detail the list of factors from each block (personal factors, job characteristics, and psychosocial factors) that are to be assessed and their justification (to prove that this list is necessary and sufficient to solve the research question).
3. In the description of the sample, it is also necessary to indicate the specifics of industries (sphere) in a given province, what percentage of workers in various industries (spheres), which industries are more developed. This is necessary to understand how the study sample correlates with the general population - the population of the province.
4. Results of the study: in the table it is also necessary to present the distribution by age and work experience , not only the average age and work experience (to make it clear how many people are in which age group and group by work experience ).
It is not clear from the description of the results whether all factors were included in the analysis (including psychosocial). This requires a more detailed description. It should be indicated whether there were differences in work ability depending on the scope of work. This may explain the resulting differences in individual labor factors.
5. Discussion of the results. Because the international journal needs to make a more detailed description of the province itself in order to be able to understand for which similar regions the results can be applied. It is also necessary to add practical recommendations based on the results of the study, in more detail the limitations of the study.
6. The solution is to make it more meaningful, reflecting the results of the study.

Best regards, the reviewer

Author Response

Dear Reviewer,  

Please find enclosed our revised manuscript. We thank the reviewer for your comments. We have revised the manuscript accordingly and provide specific answers as summary of revisions to the manuscript.

  1. 1. Introduction: despite the fact that the introduction provides data on the relevance of this study, there is not enough information about what factors affect work ability (i.e., the rationale for why the authors chose these factors in their study is not sufficiently provided).

Response 1: We thank you for your comments and useful suggestion. We have revised and added information about the factors related work ability from relevant study in introduction part on page 2, line 49-52, and line 54-57, 66-82.

  1. 2. Study design. It is necessary to present in more detail the list of factors from each block (personal factors, job characteristics, and psychosocial factors) that are to be assessed and their justification. (to prove that this list is necessary and sufficient to solve the research question).

Response 2: We have revised and added more detail the list of independent factor accordingly reviewer’ suggestion on page 2; Line 93-97.

  1. 3. In the description of the sample, it is also necessary to indicate the specifics of industries (sphere) in a given province, what percentage of workers in various industries (spheres), which industries are more developed. This is necessary to understand how the study sample correlates with the general population - the population of the province.

Response 3: This study aims to investigate the work ability level and factors influencing work ability among the working age population in Singburi Province. Thus, the information regarding to participant’s occupational was conducted during this study. The results shown in the table 1 in the Currently job part. From literature review found that in 2021, SingBuri Province has a population of 205,344 of whom 176,894 (86%) participated in the working age (15 years and older). The three labor markets with highest rate of production were industrial manufacturing sector (25.75% of Gross Provincial Product: GPP), agriculture, forestry and fisheries sector (14.48% of GPP) and trade and services sector (13.44% of GPP). As a result, this study focused on the working age population as we could collect a large range of data in the province context.

  1. 4. Results of the study: in the table it is also necessary to present the distribution by age and work experience, not only the average age and work experience (to make it clear how many people are in which age group and group by work experience ). It is not clear from the description of the results whether all factors were included in the analysis (including psychosocial). This requires a more detailed description. It should be indicated whether there were differences in work ability depending on the scope of work. This may explain the resulting differences in individual labor factors.

Response 4: 1) The Table 1 have been categorized by age and work experience group (Page 3-4). 

            2) We have explained and clarified the psychosocial factors in the discussion section accordingly reviewer’ comment. (page 8; line 236-240).

  1. 5. Discussion of the results. Because the international journal needs to make a more detailed description of the province itself in order to be able to understand for which similar regions the results can be applied. It is also necessary to add practical recommendations based on the results of the study, in more detail the limitations of the study.

Response 5: We thank you for this useful suggestion.

 1) We made changes in discussion part to address this comment by adding WAI among Asian countries: in Philippines, Malaysia Indonesia and Vietnam (page 7, line 178-179).

 2) We have more described the limitations of the study (page 8; line 246-254).

  1. 6. The solution is to make it more meaningful, reflecting the results of the study.

Response 6: We thank you for your comments and useful suggestion. we clarify in the discussion section accordingly your comments. Please find the revise version in the red color part.  

your sincerely,

Authors

Reviewer 3 Report

The article “Factors influencing work ability among the population in Singburi Province, Thailand” is interesting however, it needs some correctness:

1) In my opinion, it would be advisable if the authors prepared a separate section devoted to the subject of sustainable development in human capital, especially about improving human conditions at work. I propose to prepare a separate section on this topic, right after the introduction part.

2) In the second part entitled “Material and methods” (table 1) ought to be put information about the character of these people’s work (physical or mental work) or even the type of their position in the company. Also, it would be required what kind of company it was (public or private).

3) Authors ought to present their hypothesis, which they had to formulate before starting the research. Clear aims of this research and the whole article are also needed.

4) In par. 206-207 there has been written: “It means that as workers’ ages rise, their work ability decreases”. I do not agree with this thesis, because it depends on their work’ character. If it is mental work and high position – it is often the other way around.

5) In the Discussion section authors give the examples of similar research in Spain, Poland, Iran, Luxembourg, Cyprus. In my opinion, it is not comparable, because in those countries are extremally another circumstances as in Thailand. Such as comparative analysis of research on the same topic can be done in similar economies, like in for example other Asian countries with similar levels of socio-economic development.

6) In the Conclusion part of the article the authors should formulate some recommendations coming from their research, especially some policy recommendations which are not clear in this article. The conclusion part is very poor and requires fundamental improvement.

Author Response

Dear editor,

Please find enclosed our revised manuscript. We thank the two reviewers for their comments. We have revised the manuscript accordingly and provide specific answers in red highlight for reviewer 1 and green highlight for reviewer 2 

your sincerely

the authors

Round 2

Reviewer 1 Report

Thanks for making efforts to revised manuscript. 

Author Response

Dear Reviewer,  

Please find enclosed our revised manuscript. We thank the reviewer for your comments. We have revised the sentence accordingly in red highlight.

You sincerely,

The authors

Reviewer 2 Report

Dear authors, thank you for your additions. The article can be recommended for publication.
Best regards, the reviewer

Author Response

(The authors gave the same response as above.)
